# Association of Sodium, Potassium and Sodium-to-Potassium Ratio with Urine Albumin Excretion among the General Chinese Population

**DOI:** 10.3390/nu13103456

**Published:** 2021-09-29

**Authors:** Yuewen Sun, Puhong Zhang, Yuan Li, Feng J. He, Jing Wu, Jianwei Xu, Xiaochang Zhang, Xian Li, Jing Song

**Affiliations:** 1The George Institute for Global Health at Peking University Health Science Centre, Beijing 100600, China; ysun2@georgeinstitute.org.cn (Y.S.); yli@georgeinstitute.org.cn (Y.L.); lxian@georgeinstitute.org.cn (X.L.); 2Faculty of Medicine, University of New South Wales, Sydney, NSW 2052, Australia; 3Wolfson Institute of Preventive Medicine, Barts and The London School of Medicine & Dentistry, Queen Mary University of London, London EC1M 6BQ, UK; f.he@qmul.ac.uk (F.J.H.); jing.song@qmul.ac.uk (J.S.); 4The National Center for Chronic and Noncommunicable Disease Control and Prevention, The Chinese Center for Disease Control and Prevention, Beijing 100000, China; xujianwei@ncncd.chinacdc.cn; 5Noncommunicable Disease and Aging Health Management Division, Chinese Center for Disease Control and Prevention, Beijing 100000, China; zhangxc@chinacdc.cn

**Keywords:** albuminuria, sodium, potassium, sodium-to-potassium ratio, kidney function

## Abstract

Mixed evidence was published regarding the association of sodium, potassium and sodium-to-potassium ratio (Na/K ratio) with renal function impairment. This study was conducted to further explore the relationship between sodium, potassium, NA/K ratio and kidney function in the general adult Chinese population. We performed a cross-sectional analysis using the baseline data from the Action on Salt China (ASC) study. 5185 eligible general adult participants from the baseline investigation of the ASC study were included in this analysis. Sodium, potassium and albumin excretion were examined from 24-h urine collection. Albuminuria was defined as albumin excretion rate (AER) greater than or equal to 30 mg/24-h. Mixed linear regression models, adjusted for confounders, were fitted to analyze the association between sodium, potassium and Na/K ratio, and natural log transformed AER. Mixed effects logistic regression models were performed to analyze the odds ratio of albuminuria at each quintile of sodium, potassium and Na/K ratio. The mean age of the participants was 49.5 ± 12.8 years, and 48.2% were male. The proportion of albuminuria was 7.5%.The adjusted mixed linear models indicated that sodium and Na/K ratio was positively associated with natural log transformed AER (Sodium: β = 0.069, 95%CI [0.050, 0.087], *p* < 0.001; Na/K ratio: β = 0.026, 95%CI [0.012, 0.040], *p* < 0.001). Mixed effects logistic regression models showed that the odds of albuminuria significantly increased with the quintiles of sodium (*p* < 0.001) and Na/K ratio (*p* = 0.001). No significant association was found between potassium and the outcome indicators. Higher sodium intake and higher Na/K ratio are associated with early renal function impairment, while potassium intake was not associated with kidney function measured by albumin excretion.

## 1. Introduction

Chronic kidney disease (CKD) has become an important public health problem in China [1,2,3]. Urinary albumin excretion is an important indicator for early renal function impairment and has proved to be a strong risk factor for cardiovascular diseases [4,5,6]. Therefore, identifying risk factors related to urinary albumin excretion is essential for the primary prevention of CKD.

The association between sodium and potassium intake and kidney function has been extensively studied; however, the results were inconsistent [7,8,9,10,11,12,13,14,15]. The Prevention of Renal and Vascular End Stage Disease (PREVEND) study and the Reasons for Geographic and Racial Differences in Stroke (REGARDS) study presented that sodium intake was positively associated with albuminuria [7,8,9,10]. In comparison, some other cross-sectional studies indicated no significant association between sodium and albuminuria [11,12]. A study in the US reported that higher sodium intake was related to lower odds of CKDs [13], however, there are plenty of studies showing the opposite [7,8,9,10,14]. Similar heterogeneous findings were also reported on the association between potassium intake and kidney function [11,15,16].

Even though many studies have been carried out examining the association of sodium, potassium and kidney function, many of them have some methodological limitations. First, many studies used subjective dietary assessment methods, including food frequency questionnaires and diet recall methods, or spot urine samples to estimate 24-h sodium and potassium intake, which are not reliable and prone to recall bias [13,17,18,19,20,21]. Second, 24-h urine albumin excretion rate (AER) is the gold method for detecting albuminuria [22]. Due to the inconvenience of 24-h urine collection, previous studies often use albumin/creatine ratio (ACR) from spot urine samples as the indicator for screening albuminuria [12,16]. However, evidence proves that ACR would underestimate AER and “prone to bias introduced by differences in muscle mass” [22,23].

Using the baseline data collected from the Action on Salt China (ASC) study, this cross-sectional study aimed to examine the association of the urinary sodium, potassium excretion and sodium-to-potassium ratio (Na/K ratio) with early kidney function impairment measured by 24-h AER and odds of albuminuria. We hypothesized that high sodium, low potassium and high Na/K ratio were associated with high 24-h AER and increased odds of albuminuria.

## 2. Materials and Methods

### 2.1. Participants

The ASC study consisted of three cluster randomized controlled trials (RCTs) conducted in 6 provinces in China, representing a wide range of geographical locations, socioeconomic status and covering urban and rural residents [24,25,26,27]. All these sub-studies used consistent recruitment criteria: aged 18 to 75 years old; would not move within two years; able to collect 24-h urine. Individuals who were unable or illegible to collect 24-h urine due to pregnancy, lactation, urinary tract infection or other conditions were excluded. Participant recruitment and baseline data collection were performed from September to December 2018.

Participants whose urine collections were deemed unqualified were excluded from this analysis. The exclusion criteria are as follows: urine volume < 500 mL, or creatinine < 4.0 mmol for women or <6.0 mmol for men, or the collection duration was less than 20 h or more than 28 h. In addition, as the objective of this study is to analyze the association between sodium, potassium and early kidney function impairment, we excluded the participants whose AER was higher than 300 mg/24-h and those who reported a diagnosed history of chronic kidney diseases. Albuminuria was diagnosed if the albumin excretion rate was in the range of 30~300 mg/24-h [4].

### 2.2. Data Collection

Data collection was performed by trained researchers using an electronic data capture system. Participants underwent an interviewer-administered questionnaire, which covered the following information: (1) basic demographics (age, gender, education level, income level, marriage status and medical insurance.); (2) knowledge, attitudes and behaviors related to salt intake; and (3) existing chronic conditions, including hypertension, diabetes, chronic kidney disease, cerebral apoplexy, coronary heart disease, etc. Participants had their height, weight, waist circumference and blood pressure measured by trained researchers using calibrated devices. Smoking was a binary variable whether the participants ever smoked during the past 30 days. Alcohol intake was a categorical variable of three levels: never, occasionally and frequently. Physical activity was a binary variable whether the participants were engaged in moderate physical activity for at least 30 min and at least 3 times a week. Hypertension was diagnosed if the participant reported taking antihypertensive medication during the past two weeks, or the average of the last two blood pressure measurements was a systolic blood pressure higher than 140 mmHg or a diastolic blood pressure higher than 90 mmHg. Body mass index (BMI) was calculated as the weight in kilograms divided by the square of height in meters. Region was classified as North or South according to the geographical location of the study sites.

All participants completed timed urine collections following the instructions by trained researchers. Participants in the AIS study provided two consecutive 24-h urine collections; other participants only provided one 24-h urine collection. The starting time and ending time of urine collection were recorded by the electronic data collection system, which was then used to calculate the urine collection duration and to estimate the adjusted for 24-h urine volume. The primary independent variables are 24-h sodium excretion and 24-h potassium excretion calculated by multiplying their concentration in the collected urine samples and the adjusted volume of urine collected for 24 h.

### 2.3. Statistical Analysis

Continuous data were presented as mean with standard deviation, and categorical data as frequencies and percentages. Demographic, clinical and socioeconomic variables were compared across subjects with and without albuminuria using Chi-squared test or t test. We described the distribution of AER using box plots by each quintile of 24-h sodium excretion, 24-h potassium excretion and Na/K ratio, respectively. The Kruskal-Wallis test was to examine whether the distribution of AER was different between each comparison pair of quintile groups.

Linear mixed models were used to examine the association between the level of 24-h sodium excretion, 24-h potassium excretion, Na/K ratio and AER level. Due to the skewed distribution of AER, we performed a natural log transformation of it before fitting the models. We accounted for the possible heterogeneity across the 3 RCTs and the potential aggregation of participants within the same community by specifying the random effects of RCT and community levels in the mixed models. We also adjusted potential confounders in the mixed models. Model 1 assessed the effect of 24-h sodium excretion and 24-h potassium excretion or Na/K ratio without adjusting covariates. Model 2 adjusted age and gender. Based on Model 2, Model 3 further adjusted region (north/south), smoking, alcohol intake, physical activity, BMI, diabetes, systolic blood pressure and antihypertensive medication use.

Mixed effects logistic regression was performed to examine the association between the quintiles of 24-h sodium excretion, 24-h potassium excretion and Na/K ratio and proportion of albuminuria, with the lowest quintile as the reference group. Model 1 assessed the association between the quintiles of 24-h sodium and potassium excretion or the quintiles of Na/K ratio and the presence of albuminuria without adjusting covariates. Model 2 adjusted age and gender. Based on Model 2, Model 3 further adjusted region (north/south), smoking, alcohol intake, physical activity, BMI, diabetes, systolic blood pressure and antihypertensive medication use. In addition, the random effects of RCT and community levels were specified in all models.

As sensitivity analysis, we conducted similar linear mixed models and mixed effects logistic regression models using all urine samples regardless of whether they met the urine sample quality criteria. All *p* values were two-sided and used *p* below 0.05 as significant. SAS Enterprise Guide 8.3 was used for statistical analysis. Stata 14.2 was used for making figures.

## 3. Results

In total, 5185 participants were included in this analysis after excluding 268 participants according to the exclusion criteria. Details were shown in the flow chart of participants (Figure 1).

The mean age of the participants was 49.5 ± 12.8 years, and 48.2% were male. The median of AER was 3.94 mg/24-h (interquartile range: 2.52, 7.26). The proportion of albuminuria among the eligible participants was 7.5%. Baseline characteristics of the participants were compared by the presence of albuminuria and shown in Table 1. Compared with those without albuminuria, participants with albuminuria were more likely to be older, be male, live in the south, be smokers, have higher BMI and higher sodium intake and Na/K ratio, have a higher prevalence of hypertension, diabetes and obesity.

### 3.1. AER Levels across the Quintiles of Sodium, Potassium and Na/K Ratio

The intervals for quintiles 1–5 of 24-h sodium excretion were ≤2.83, 2.84~3.66, 3.67~4.50, 4.51~5.65, >5.65 g/24-h, respectively; quintiles of potassium excretion 1–5 were ≤1.09, 1.10~1.35, 1.36~1.61, 1.62~1.99, >1.99 g/24-h, respectively; quintiles of Na/K ratio were ≤3.33, 3.34~4.22, 4.23~5.15, 5.16~6.46, >6.46, respectively.

Figure 2 is a box-whisker plot of AER across the quintiles of 24-h sodium excretion, 24-h potassium excretion and Na/K ratio, respectively. It showed that participants in higher quintiles of 24-h sodium excretion and higher Na/K ratio would excrete more albumin, and the increasing trend was evident. In comparison, the trend was not so evident between the potassium quintiles and AER. All pairs of quintile groups that exhibited significant difference in the Kruskal Wallis test were showed by connected lines in Figure 2. The distribution of AER was significantly different between the following sodium quintile pairs: Q1 vs. Q3, Q1 vs. Q4, Q1 vs. Q5, Q2 vs. Q4, Q2 vs. Q5, Q3 vs. Q4, Q3 vs. Q5. In comparison, less pairs of potassium quintiles and Na/K ratio quintiles showed significant difference in the distribution of AER.

### 3.2. Adjusted Association between the Levels of Sodium, Potassium, Na/K Ratio and AER Level

Table 2 presented the results from the linear mixed models showing the association between the level of sodium, potassium, Na/K ratio and natural log transformed AER. All three models showed that the association of sodium and Na/K ratio with natural log transformed AER was significant and positive. In the fully adjusted model, the coefficient between sodium and log transformed AER was 0.069 (95%CI: 0.050, 0.087) and the coefficient between Na/K ratio and log transformed AER was 0.026 (95%CI: 0.012, 0.040). These results indicated that sodium and Na/K ratio were significantly associated with AER. After adjusting all covariates, every 1000 mg increase of sodium excretion would cause a rise of AER by 7.14% (95%CI: 5.13%, 9.09%). Every unit increase of Na/K ratio would cause an increase of AER by 2.63% (95%CI: 1.21%, 4.08%). For potassium, we did not find any significant association of it with AER in any models.

### 3.3. Adjusted Association between the Quintiles of Sodium, Potassium, Na/K Ratio and Proportions of Albuminuria

Figure 3A–C presented the main results of the mixed effects logistic regression models. The covariates-adjusted odds ratio (OR) and 95% confidence interval (95%CI) of albuminuria across the quintile intervals of sodium, potassium and Na/K ratio used the first quintile as the reference category. In model 1, when no covariates were adjusted, the third to fifth sodium quintiles were significantly associated with the increased odds of albuminuria compared with the lowest quintile. After adjustment for confounders in model 2 and model 3, the ORs of all quintiles decreased slightly, and the fourth and fifth quintiles of sodium remained significantly associated with higher odds of albuminuria. In model 3, the OR of the fifth quintiles of sodium was 2.154 (95%CI: 1.431, 3.242). In all models, the ORs for the potassium quintiles were primarily insignificant. The fourth and fifth quintiles quintile of the Na/K ratio were significantly associated with increased odds of albuminuria in all three models. In model 3, the OR of the fifth quintiles of Na/K ratio was 1.953 (95%CI: 1.356, 2.813). The *p* value in Figure 3A–C indicated whether there was a statistically significant association between the quintiles of sodium, potassium and Na/K ratio and the proportion of albuminuria after adjusting for covariates in the logistic regression models. As showed in this figure, quintiles of sodium and Na/K ratio were significantly associated with the proportion of albuminuria in all three models, while quintiles of potassium were not found associated with the outcome variable in any models. Detailed results of the mixed effects logistic regression models were showed in the Appendix A.

### 3.4. Sensitivity Analysis

All urine samples were included in the sensitivity analysis regardless of whether they met the urine sample quality criteria. In total, 5275 participants were included in sensitivity analyses. The findings were similar in the linear mixed models and mixed effects logistic regression models. Detailed results of the sensitivity analysis are shown in Table 3 and Table 4.

## 4. Discussion

This study is the largest to date in measuring the association between sodium, potassium, Na/K ratio and kidney function impairment using 24-h urine collection among the general Chinese adult population. We found that higher sodium excretion and higher Na/K ratio were associated with increased 24-h AER and higher odds of albuminuria after adjustment for potential confounders, including sex, age, region, smoking, alcohol intake, physical activity, BMI, diabetes, systolic blood pressure and antihypertensive medication use. These findings highlight the importance of high sodium in kidney function impairment, and potential chronic kidney diseases. As for potassium, we could not find any association of it with 24-h AER and the presence of albuminuria.

The adverse effect of high sodium intake on kidney function has been reported in a few studies [8,9,19,28,29,30]. The PREVEND study conducted in the Netherlands found a positive relation between 24-h sodium excretion and urine albumin excretion [30]. In this study, the researchers found that each 1-g increase of sodium intake was associated with 4.6 mg/d increase in urinary albumin excretion after adjustment for confounders [30], which is higher than our findings that 1-g increase of sodium would increase the log AER by 0.069. The difference is possibly due to the varied AER level among the studied population: the median of AER among the PREVEND population was about 8 mg/d [30], which is much higher than the 3.94 mg/d of our studied population. A cross-section study in China found that the odds of albuminuria increased by 174% in the 4th quartile of 24-h sodium excretion after adjusting for age, gender, smoking, alcohol consumption, BMI, hypertension and diabetes [7]. A cross-sectional study on the Korean adult population found that the fourth quartile of estimated 24-h sodium excretion was significantly associated with 60% increased odds of albuminuria after adjusting for age, sex, diabetes, obesity and hypertension [28]. A longitudinal cohort study found that 24-h sodium excretion and Na/K ratio were predictors of kidney function decline measured by estimated glomerular filtration rate(eGFR) [9]. Another two cohort studies conducted among the Asian population showed that higher 24-h Na/K ratio was associated with higher rate of CKD progression measured by 50% decrease of eGFR or the onset of end-stage renal disease after 5-year or 6-year follow-up [31,32]. Even though these studies used a different indicator of kidney function, their findings were similar with ours which indicates that the association between high sodium and kidney function impairment is robust.

Furthermore, two randomized controlled trials suggested the causal association between high dietary salt intake and early renal function impairment and demonstrated that sodium reduction could reduce albumin excretion and protect kidney function. A cluster-randomized trial for salt reduction in China found that a 0.82 g/day dietary sodium reduction could reduce albuminuria risk by 33% [8]. Another salt reduction trial found that 24-h albumin excretion decreased about 10% among the participants after 6-week low salt intervention [33].

Our study showed that potassium intake was not associated with 24-h AER or albuminuria. Several population-based studies have reported similar findings [7,9,20,29]. However, other researchers reported that potassium had protective effects on kidney function, and higher potassium intake could reduce albumin excretion [12,13,16]. For example, the Coronary Artery Risk Development In young Adults (CARDIA) study among younger adults aged 23–35 found that every 1000 mg/day increase of 24-h potassium excretion was associated with 29% decreased risk of incident albuminuria [16]. In addition, a RCT conducted among mild hypertensives showed that higher potassium intake could reduce 24-h AER and ACR [34]. These evidence indicated that the association between potassium intake and kidney function was possible. However, such an inverse association between 24-h potassium excretion and AER or albuminuria has not been found in previous Chinese studies [7,8], as well as our study. Previous studies reported that potassium intake in China is below the recommended amount [35,36]. Among our surveyed participants, the mean daily potassium intake was 1.57 g (SD = 0.61), and the interquartile range was 1.16~1.88 g. The potassium intake of nearly all participants in this study was below the lower limit of 3.5 g per day recommended by the World Health Organization [37]. Therefore, although our analysis did not find the association between potassium and AER and albuminuria, we cannot extrapolate our findings to a broader and higher amount of potassium intake.

The biological mechanism of sodium and potassium on kidney function is still unclear. Some researchers believed the effect of sodium and potassium on kidney function is mediated through blood pressure [38]. The increased blood pressure caused by higher sodium intake could deteriorate kidney function, while potassium could be kidney protective as it could lower blood pressure. However, some researchers believed that sodium might impair kidney function, even in the absence of a change in blood pressure. In addition, there were two theories. First, higher sodium intake might cause inflammation and therefore damage kidney function [39,40]. Second, high sodium intake would induce impairments in endothelial function, and potassium would counteract such deleterious effects [41]. Both theories need further investigations.

### Strengths and Weaknesses of This Study

In this study, we recruited a large sample from 6 provinces, which could well represent the adult Chinese population of different socio-economic backgrounds and dietary habits. We collected 24-h urine samples, which could provide reliable data for measuring sodium, potassium and albumin excretion. The use of 24-h AER would improve the accuracy of detecting albuminuria and screening early kidney function impairment.

There are several limitations we need to acknowledge. First, we did not collect blood samples; therefore, we could not measure kidney function using the estimated glomerular filtration rate. Second, we should be cautious when interpreting the findings due to the nature of the cross-sectional study and could not draw causal inferences from this study. Third, this study and many of the studies mentioned above were secondary analyses of existing databases from large cohort studies and population-based surveys [13,19,20,42]. Therefore, only available data regarding potential confounders could be adjusted in the analysis. Due to the complexity of kidney function impairment, residual confounding is still possible.

## 5. Conclusions

Our study indicated that high sodium excretion and Na/K ratio are significantly associated with the development of renal function impairment, measured by 24-h AER and the presence of albuminuria. Thus, this study indicates that high sodium intake would play an important role in kidney function decline. In addition, the sodium reduction initiatives might have beneficial impacts on kidney function protection.

## Figures and Tables

**Figure 1 nutrients-13-03456-f001:**
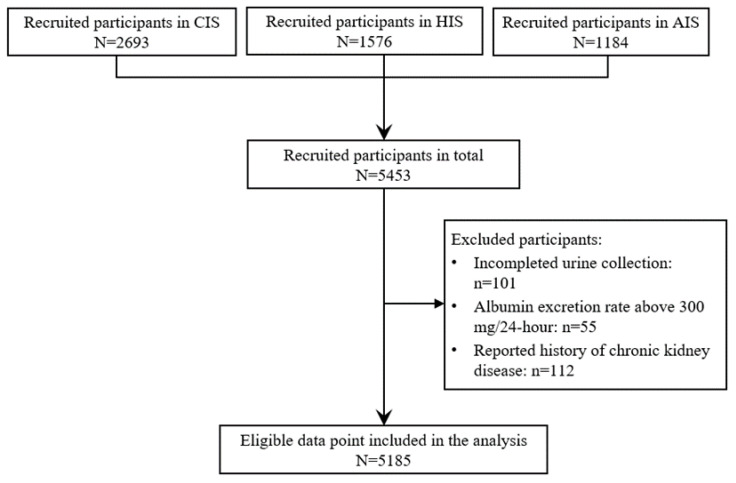
Flowchart of participants. Notes: Community-based comprehensive salt reduction intervention study (CIS), Home cook salt reduction intervention study (HIS) and App-based salt reduction in primary school children and their families (AIS) are the 3 independent trials in Action on Salt China program.

**Figure 2 nutrients-13-03456-f002:**
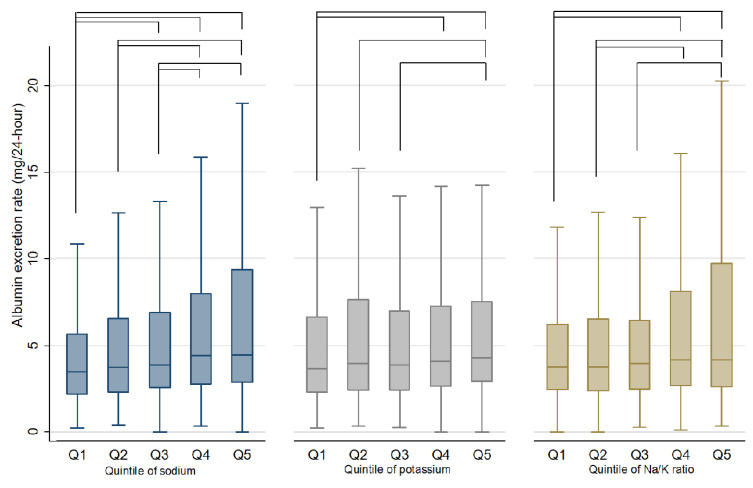
Box-and-whisker plots of 24-h urinary albumin excretion rate for the quintiles of sodium, potassium and sodium-to-potassium ratio. Notes: Q indicates quintile. The lines in the figure connected the pairs that exhibited significant difference for the distribution of AER in the Kruskal Wallis test. The adjusted *p* value for significance of multiple comparison between quintile groups was 0.0025. The pairs of significant difference were the follows. Sodium quintile: Q1 vs. Q3, Q1 vs. Q4, Q1 vs. Q5, Q2 vs. Q4, Q2 vs. Q5, Q3 vs. Q4, Q3 vs. Q5. Potassium quintile: Q1 vs. Q4, Q1 vs. Q5, Q2 vs. Q5, Q3 vs. Q5. Na/K ratio quintile: Q1 vs. Q4, Q1 vs. Q5, Q2 vs. Q4, Q2 vs. Q5, Q3 vs. Q5.

**Figure 3 nutrients-13-03456-f003:**
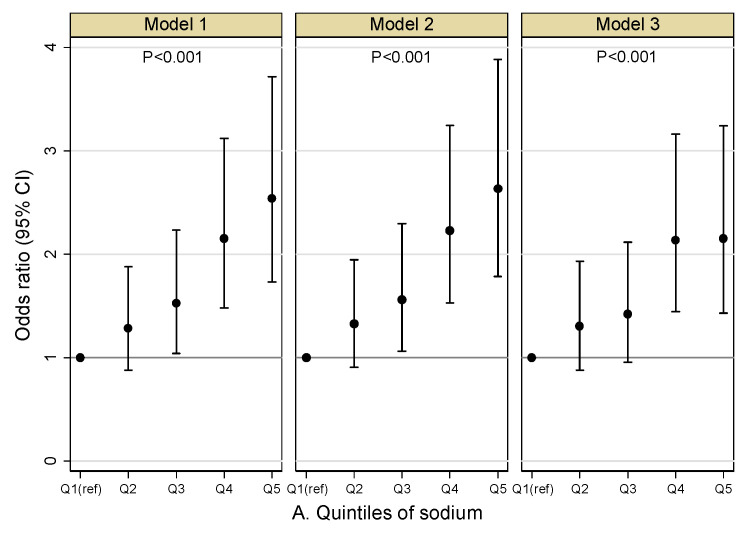
(**A**–**C**) Odds ratio and 95% confidence interval of albuminuria at each quintile of urinary index. Notes: The first quintile was used as the reference level in the mixed effects logistic regression models. Model 1 examined the association of sodium and potassium or Na/K ratio quintiles with the odds of albuminuria without adjustment. Model 2 adjusted age and sex; model 3 adjust age, sex, region (north/south), smoking, alcohol intake, physical activity, BMI, diabetes, systolic blood pressure and antihypertensive medication use.

**Table 1 nutrients-13-03456-t001:** Characteristics of participants in the ASC study by the presence of albuminuria.

Variables	Overall (*n* = 5185)	Albuminuria		*p*
		No (*n* = 4798)	Yes (*n* = 387)	
Age (years), mean ± SD	49.5 ± 12.8	49.2 ± 12.8	52.9 ± 12.1	<0.001
Gender: male, *n* (%)	2500 (48.2)	2279 (47.5)	221 (57.1)	<0.001
Region, *n* (%)				
North	2419 (46.7)	2263 (47.2)	156 (40.3)	0.009
South	2766 (53.4)	2535 (52.8)	231 (59.7)	
AER (mg/24 h), median (IQR)	3.9 (2.5~7.3)	3.7 (2.4~5.9)	57.6 (39.4~99.4)	<0.001
Sodium (g/24 h), mean ± SD	4.3 ± 1.8	4.3 ± 1.8	4.8 ± 2.0	<0.001
Potassium (g/24 h)	1.6 ± 0.6	1.6 ± 0.6	1.6 ± 0.6	0.993
Na/K ratio, mean ± SD	5.0 ± 2.1	5.0 ± 2.1	5.6 ± 2.3	<0.001
SBP (mmHg), mean ± SD	125.2 ± 19.0	124.2 ± 18.4	137.9 ± 21.4	<0.001
DBP (mmHg), mean ± SD	79.0 ± 11.1	78.5 ± 10.8	85.5 ± 12.7	<0.001
Waist circumference (cm), mean ± SD	84.7 ± 10.6	84.3 ± 10.5	89.6 ± 10.5	<0.001
BMI, *n* (%)				
<24	1982 (38.2)	1900 (39.6)	82 (21.2)	<0.001
24–28	2086 (40.2)	1923 (40.1)	163 (42.1)	
≥28	1117 (21.5)	975 (20.3)	142 (36.7)	
Education years, *n* (%)				
0–6	2047 (39.5)	1882 (39.2)	165 (42.7)	0.156
7–9	1749 (33.7)	1615 (33.7)	134 (34.6)	
≥10	1389 (26.8)	1301 (27.1)	88 (22.7)	
Smoking: yes, *n* (%)	1395 (26.9)	1273 (26.5)	122 (31.5)	0.033
Alcohol intake, *n* (%)				
No	3061 (59.0)	2836 (59.1)	225 (58.1)	0.606
Occasionally	1622 (31.3)	1503 (31.3)	119 (30.7)	
Frequently	501 (9.7)	458 (9.6)	43 (11.1)	
Physical activity: active, *n* (%)	1618 (31.2)	1480 (30.8)	138 (35.7)	0.050
High blood pressure, *n* (%)	1662 (32.1)	1424 (29.7)	238 (61.5)	<0.001
Antihypertensive medication: yes, *n* (%)	801 (15.4)	673 (14.0)	128 (33.1)	<0.001
Diabetes, *n* (%)	282 (5.4)	221 (4.6)	61 (15.8)	<0.001

Note: SD indicates standard deviation; AER, albumin excretion rate; IQR, interquartile range; BMI, body mass index; SBP, Systolic blood pressure; DBP, diastolic blood pressure.

**Table 2 nutrients-13-03456-t002:** Association of urinary index (g/d) and natural log transformed albumin excretion rate (mg/d) using the mixed linear effect models.

Models	β	95%CI	*p*
Sodium			
Model 1	0.090	0.071~0.108	<0.001
Model 2	0.095	0.077~0.114	<0.001
Model 3	0.069	0.050~0.087	<0.001
Potassium			
Model 1	0.011	−0.041~0.063	0.679
Model 2	0.001	−0.051~0.053	0.977
Model 3	0.039	−0.012~0.089	0.135
Na/K ratio			
Model 1	0.041	0.027~0.055	<0.001
Model 2	0.045	0.031~0.059	<0.001
Model 3	0.026	0.012~0.040	<0.001

Note: Model 1 examined the association of sodium and potassium or Na/K ratio with natural logarithmic transformed AER without adjustment. Model 2 adjusted age and sex; model 3 adjust age, sex, region (north/south), smoking, alcohol intake, physical activity, BMI, diabetes, systolic blood pressure and antihypertensive medication use.

**Table 3 nutrients-13-03456-t003:** Sensitivity analysis-Association of urinary index and urine albumin excretion rate (natural logarithmic transformed) using the mixed linear model.

Models	β	95%CI	*p*
Sodium			
Model 1	0.092	0.074~0.110	<0.001
Model 2	0.098	0.080~0.117	<0.001
Model 3	0.071	0.053~0.089	<0.001
Potassium			
Model 1	0.023	−0.029~0.074	0.386
Model 2	0.012	−0.040~0.063	0.661
Model 3	0.045	−0.006~0.095	0.081
Na/K ratio			
Model 1	0.035	0.022~0.048	<0.001
Model 2	0.039	0.025~0.052	<0.001
Model 3	0.022	0.009~0.035	0.001

Note: Model 1 examined the association of sodium and potassium or Na/K ratio with natural logarithmic transformed AER without adjustment. Model 2 adjusted age and sex; model 3 adjust age, sex, region (north/south), smoking, alcohol intake, physical activity, BMI, diabetes, systolic blood pressure and antihypertensive medication use.

**Table 4 nutrients-13-03456-t004:** Sensitivity analysis-Odds ratio (OR) and 95% confidence interval of albuminuria at each quintile of urinary index using mixed effects logistic regression model.

Factors	Model 1			Model 2			Model 3		
	OR	95%CI	*p*	OR	95%CI	*p*	OR	95%CI	*p*
Sodium (g/24 h)									
Q1 (≤2.78)	Ref		<0.001	Ref		<0.001	Ref		0.001
Q2 (2.79~3.62)	1.288	0.882~1.880		1.339	0.915~1.959		1.306	0.882~1.934	
Q3 (3.63~4.48)	1.549	1.062~2.260		1.593	1.089~2.330		1.440	0.972~2.132	
Q4 (4.49~5.63)	2.089	1.439~3.031		2.167	1.487~3.158		2.058	1.391~3.043	
Q5 (>5.63)	2.460	1.680~3.602		2.564	1.739~3.780		2.080	1.382~3.131	
Potassium (g/24 h)									
Q1 (≤1.07)	Ref		0.094	Ref		0.106	Ref		0.215
Q2 (1.08~1.34)	1.011	0.725~1.410		1.000	0.715~1.397		0.998	0.705~1.413	
Q3 (1.35~1.60)	0.678	0.473~0.972		0.681	0.474~0.980		0.691	0.474~1.007	
Q4 (1.61~1.98)	0.795	0.558~1.132		0.800	0.560~1.143		0.812	0.560~1.178	
Q5 (>1.98)	0.735	0.510~1.058		0.721	0.499~1.042		0.769	0.521~1.133	
Na/K ratio									
Q1 (≤3.33)	Ref		<0.001	Ref		<0.001	Ref		0.002
Q2 (3.34~4.22)	1.185	0.818~1.717		1.213	0.835~1.762		1.144	0.777~1.683	
Q3 (4.23~5.15)	1.254	0.868~1.811		1.280	0.884~1.853		1.204	0.820~1.768	
Q4 (5.16~6.46)	1.619	1.140~2.300		1.639	1.150~2.337		1.595	1.103~2.305	
Q5 (>6.46)	2.191	1.563~3.072		2.271	1.611~3.204		1.915	1.332~2.752	

Note: Q indicates quintile. Model 1 examined the association of sodium and potassium or Na/K ratio quintiles with the odds of albuminuria without adjustment. Model 2 adjusted age and sex; model 3 adjust age, sex, region (north/south), smoking, alcohol intake, physical activity, BMI, diabetes, systolic blood pressure and antihypertensive medication use.

## Data Availability

The datasets used and/or analyzed during the current study are available from the corresponding authors on reasonable request.

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
