# Peer review of "Association of Sodium, Potassium and Sodium-to-Potassium Ratio with Urine Albumin Excretion among the General Chinese Population"

_nutrients, 2021, doi:10.3390/nu13103456_

Round 1

Reviewer 1 Report

The article entitled “Association of sodium, potassium, and sodium-to-potassium ratio with urine albumin excretion among the general Chinese  population” is a well written study however it contains some flaws that should be improved.

 I would like to point out some issues that should be revised by authors before publication.

Line 47 – The authors mention that “The association between sodium and potassium intake and kidney function has been extensively studied; however, the results were inconsistent.”, however there is no references in this part. I suggest to include the references in the end of the sentence.

Line 52 – The authors mention that “A study in the US reported that higher sodium intake was related to lower odds of CKD”, however, there is plenty of studies showing the opposite (i.e. Hendriksen MAH, Over EAB, Navis G, Joles JA, Hoorn EJ, Gansevoort RT, Boshuizen HC. Limited salt consumption reduces the incidence of chronic kidney disease: a modeling study. J Public Health (Oxf). 2018 Sep 1;40(3):e351-e358. doi: 10.1093/pubmed/fdx178.). This part should be rephrased, emphasizing that this topic is still controversial.

Line 84 - The verbal tense in methods should be in the past.

Line 93 – It is not clear to the reader how this information was processed or computed “2) knowledge, attitudes, and behaviors related to salt intake;”. Which questions were made? Was it a specific and valid questionnaire? Authors should improve this.

Line 96 – “waist circumstance” should be waist circumference.

Line 97 – “drinking” should be described as alcohol intake.

Line 119 – please revise the statistical tests’ names.

Line 152 – There is no need of this sentence since it was already describe in methods section: “Incomplete urine collection is defined as follows: urine volume < 500 ml 152 or creatinine < 4.0 mmol for women or < 6.0 mmol for men or the collection duration was 153 less than 20 hours or more than 28 hours.”

Line 159 – Revise English writing;

Line 159 - table should be written with upper case.

Line 160 – The results should not be described as “were more likely to be”, please rephrase.

Table 1. – Please use p<0.001 instead of 0.000

              – Please use alcohol intake instead of drinking

Line 175 – “the trend was statistically significant”, was it a trend or a statistically significant? If the latter, the p value should be described.

Figure 2. – Again, it is not clear for the reader where is the statistically significance mentioned by the authors in the line 175. Please use symbols in the figure and a proper footnote to better describe it.

Table 2. – Again, please use p<0.001 instead of 0.000.

Figure 3A-3C. – Please use symbols and footnote to describe the statistically significance between the quintiles.

Discussion is well written.

Author Response

Dear Reviewer, 

Thanks a lot for spending time reviewing our manuscript. We appreciate the time and effort that you dedicated to providing feedback on our manuscript and are grateful for the insightful comments on and valuable improvements to our paper.
We have incorporated most of the suggestions. Those changes are highlighted
using tracked changes within the manuscript. Please see below, for a point-by-point response to the reviewer’s comments. 

Point 1: Line 47 – The authors mention that “The association between sodium and potassium intake and kidney function has been extensively studied; however, the results were inconsistent.”, however there is no references in this part. I suggest to include the references in the end of the sentence.

 Response 1: Thanks for your suggestion. The references were added in the end of the sentence.

Point 2: Line 52 – The authors mention that “A study in the US reported that higher sodium intake was related to lower odds of CKD”, however, there is plenty of studies showing the opposite (i.e. Hendriksen MAH, Over EAB, Navis G, Joles JA, Hoorn EJ, Gansevoort RT, Boshuizen HC. Limited salt consumption reduces the incidence of chronic kidney disease: a modeling study. J Public Health (Oxf). 2018 Sep 1;40(3):e351-e358. doi: 10.1093/pubmed/fdx178.). This part should be rephrased, emphasizing that this topic is still controversial.

Response 2: One sentence of paraphrase was added in Line 54.

Point 3: Line 84 - The verbal tense in methods should be in the past.

Response 3: Thanks for your suggestion. The verbal tense was checked and revised.

Point 4: Line 93 – It is not clear to the reader how this information was processed or computed “2) knowledge, attitudes, and behaviors related to salt intake;”. Which questions were made? Was it a specific and valid questionnaire? Authors should improve this.

Response 4: As the KAP questionnaire was not the core of this study, we did not emphasize this part in this manuscript. We have published the protocol of ASC study. Detailed explanation of the KAP questionnaire can be found in the protocol and the reference of this protocol was added in this manuscript.

Zhang P., He F.J., Li Y., Li C., Wu J., Ma J., Zhang B., Wang H., Li Y., Han J., Luo R., He J., Li X., Liu Y., Wang C., Tan M., MacGregor G.A., Li X. Reducing Salt Intake in China with “Action on Salt China” (ASC): Protocol for Campaigns and Randomized Controlled Trials. JMIR Res Protoc. 2020;9(4):e15933.

Point 5: Line 96 – “waist circumstance” should be waist circumference.

Response 5: Thank you for pointing out this spelling mistake. The revision was made accordingly.

Point 6: Line 97 – “drinking” should be described as alcohol intake.

Response 6: Thanks for your suggestion. We replaced “drinking” by “alcohol intake” in the manuscript for clarification.

Point 7: Line 119 – please revise the statistical tests’ names.

Response 7: The statistical test used here was Chi-squared test. And the revision was made accordingly.

Point 8: Line 152 – There is no need of this sentence since it was already describe in methods section: “Incomplete urine collection is defined as follows: urine volume < 500 ml 152 or creatinine < 4.0 mmol for women or < 6.0 mmol for men or the collection duration was 153 less than 20 hours or more than 28 hours.”

Response 8: Thanks for pointing this out. This sentence was deleted.

Point 9: Line 159 – Revise English writing;

Response 9: This sentence was revised for clarification. “Baseline characteristics of the participants were compared by the presence of albuminuria and shown in Table 1.”

Point 10: Line 159 - table should be written with upper case.

Response 10: We replaced “table” with “Table” here.

Point 11: Line 160 – The results should not be described as “were more likely to be”, please rephrase.

Response 11: Thanks for your suggestion. We further clarified this comparison. “Compared with those without albuminuria, participants with albuminuria were more likely to be …”

Point 12: Table 1. – Please use p<0.001 instead of 0.000

              – Please use alcohol intake instead of drinking

Response 12: We replaced “0.000” with “<0.001” and used “alcohol intake” to replace “drinking” in Table 1.

Point 13: Line 175 – “the trend was statistically significant”, was it a trend or a statistically significant? If the latter, the p value should be described.

Response 13: The box and whisker plot were used to display the distribution of 24-hour AER across the quintiles of sodium excretion, potassium excretion and Na/K ratio. We aimed to show the trend in this plot. The sentence was revised to avoid confusion. “…, and the increasing trend was evident.”

Point 14: Figure 2. – Again, it is not clear for the reader where is the statistically significance mentioned by the authors in the line 175. Please use symbols in the figure and a proper footnote to better describe it.

Response 14: We performed nonparametric equality-of-medians test to examine whether the median of AER was equal across the quintiles of 24h sodium, potassium and Na/K ratio. And the P value showed that the medians of AER were significantly different across the quintiles of sodium, potassium, and Na/K ratio. The revision was made to figure 2 and related parts in Method and Results section.

Point 15: Table 2. – Again, please use p<0.001 instead of 0.000.

Response: 15: The revision was made accordingly.

Point 16: Figure 3A-3C. – Please use symbols and footnote to describe the statistically significance between the quintiles.

Response 16: Thanks for your suggestion. To further clarify the significance, we added the P values from the models in the Figure3A-C.

Point 17: Discussion is well written.

Response 17: Thanks.

Reviewer 2 Report

This is a very interesting manuscript. The title is very clear.

Abstract need to present p values of the regression analyses, the beta is not necessary.

Introduction is presenting the topic very well, however, I will suggest adding estimated or hypothesised results at end of introduction. Authors nee to explain that dietary and biological assessment methods are very important. Shim JS, Oh K, Kim HC. Dietary assessment methods in epidemiologic studies. Epidemiol Health. 2014 Jul 22;36:e2014009. doi: 10.4178/epih/e2014009. PMID: 25078382; PMCID: PMC4154347.

Methodology – data collection 1) basic demographics (age, gender, education level, 92 income level etc.) don’t use etc list all variables please.

Participants had their height, weight…, how and by whom?

Check line 119 do you mean Chi2 test

Line 136-138 give details about models 1,2,3.

Results change all P = 0.000 to 0.001

Results need to be unified x.xx or x.x the authors are inconsistent.

Figure 2 is useless.

Supplemental tables are important especially Table 2 - Sensitivity analysis-Association of urinary index and urine albumin excretion rate (natural logarithmic transformed) using the mixed linear model

All supplemental Tables need to be in main paper.

Discussion missed two important papers

 Koo, H., Hwang, S., Kim, T. H., Kang, S. W., Oh, K. H., Ahn, C., & Kim, Y. H. (2018). The ratio of urinary sodium and potassium and chronic kidney disease progression: Results from the KoreaN Cohort Study for Outcomes in Patients with Chronic Kidney Disease (KNOW-CKD). Medicine97(44), e12820. https://doi.org/10.1097/MD.0000000000012820

Apparently healthy urban residents with an almost within normal range mean baseline eGFR and high e24hUNa/K ratios had an increased risk for a future decline in renal function. Reducing the Na/K ratio may be important in the prevention of chronic kidney disease in its early stage

Hattori, H., Hirata, A., Kubo, S., Nishida, Y., Nozawa, M., Kawamura, K., Hirata, T., Kubota, Y., Sata, M., Kuwabara, K., Higashiyama, A., Kadota, A., Sugiyama, D., Miyamatsu, N., Miyamoto, Y., & Okamura, T. (2020). Estimated 24 h Urinary Sodium-to-Potassium Ratio Is Related to Renal Function Decline: A 6-Year Cohort Study of Japanese Urban Residents. International journal of environmental research and public health17(16), 5811. https://doi.org/10.3390/ijerph17165811

Check also this preprint from Srilanka. https://www.medrxiv.org/content/10.1101/2020.04.17.20068833v1.full

Author Response

Dear Reviewer, 

We appreciate the time and effort that you dedicated to providing feedback on our manuscript and are grateful for the insightful comments on and valuable improvements to our paper.
We have incorporated most of the suggestions. Those changes are highlighted
using tracked changes within the manuscript. Please see below, for a point-by-point response to the reviewer’s comments. 

Point 1: This is a very interesting manuscript. The title is very clear.

 Response 1: Thanks.

Point 2: Abstract need to present p values of the regression analyses, the beta is not necessary.

Response 2: Thanks for your comment. We added the P value of the regression analysis in the Abstract.

Point 3: Introduction is presenting the topic very well, however, I will suggest adding estimated or hypothesised results at end of introduction. Authors nee to explain that dietary and biological assessment methods are very important. Shim JS, Oh K, Kim HC. Dietary assessment methods in epidemiologic studies. Epidemiol Health. 2014 Jul 22;36:e2014009. doi: 10.4178/epih/e2014009. PMID: 25078382; PMCID: PMC4154347.

Response 3: The hypothesis was added at the end of introduction section. “We hypothesized that high sodium, low potassium and high Na/K ratio were associated with higher 24-hour AER and higher odds of albuminuria.” Thank you for referring this study. We added this reference in Line 61.

Point 4:  Methodology – data collection 1) basic demographics (age, gender, education level, 92 income level etc.) don’t use etc list all variables please.

Response 4: We added the full list of the basic demographic variables in the revised manuscript. “1) basic demographics (age, gender, education level, income level, marriage status and medical insurance)”

Point 5:  Participants had their height, weight…, how and by whom?

Response 5:  Thanks for your comment. Participants had their height, weight, waist circumference and blood pressure measured by trained researchers using calibrated devices. The manuscript was revised accordingly.

Point 6:  Check line 119 do you mean Chi2 test

Response 6: Yes. The statistical test used for comparison was Chi2 test. Line 119 was revised. 

Point 7:  Line 136-138 give details about models 1,2,3.

Response 7:  Thanks for your comment. We added the details regarding the models of mixed-effects logistic regression in the revised manuscript. “Model 1 assessed the association between the quintiles and the presence of albuminuria without adjusting covariates. Model 2 adjusted age and gender. Based on Model 2, Model 3 further adjusted region (north/south), smoking, alcohol intake, physical activity, BMI, diabetes, systolic blood pressure, and antihypertensive medication use.”

Point 8:  Results change all P = 0.000 to 0.001

Response 8:  Both reviewers commented on this issue. And the other reviews suggested us replacing 0.000 with <0.001. After discussion with the statistician, we replaced all 0.000 with <0.001 throughout the whole manuscript for clarification and consistency.

Point 9: Results need to be unified x.xx or x.x the authors are inconsistent.

Response 9: Thanks for your comment. All results in Table 1 were presented with one decimal for consistency.

Point 10:  Figure 2 is useless.

Response 10:  The box and whisker plot were used to display the distribution of 24-hour AER across the quintiles of sodium excretion, potassium excretion and Na/K ratio. To We performed nonparametric equality-of-medians tests to examine whether the median of AER was equal across the quintiles of 24h sodium, potassium, and Na/K ratio. And the P value showed that the medians of AER were significantly different across the quintiles of sodium, potassium, and Na/K ratio. The revision was made to figure 2 and related parts in Method and Results section.

Point 11:  Supplemental tables are important especially Table 2 - Sensitivity analysis-Association of urinary index and urine albumin excretion rate (natural logarithmic transformed) using the mixed linear model

Response 11: Thanks for your suggestion. In the revised manuscript, we added the results of sensitivity analysis (Table S2 and Table S3) in the main manuscript.

Point 12:  All supplemental Tables need to be in main paper.

Response 12: We added Table S2 and Table S3 in the main paper. As for Table S1, we kept it as a supplementary table because the information was already presented in Figure 3A-C.

Point 13:  Discussion missed two important papers

 Koo, H., Hwang, S., Kim, T. H., Kang, S. W., Oh, K. H., Ahn, C., & Kim, Y. H. (2018). The ratio of urinary sodium and potassium and chronic kidney disease progression: Results from the KoreaN Cohort Study for Outcomes in Patients with Chronic Kidney Disease (KNOW-CKD). Medicine, 97(44), e12820. https://doi.org/10.1097/MD.0000000000012820

Apparently healthy urban residents with an almost within normal range mean baseline eGFR and high e24hUNa/K ratios had an increased risk for a future decline in renal function. Reducing the Na/K ratio may be important in the prevention of chronic kidney disease in its early stage

Hattori, H., Hirata, A., Kubo, S., Nishida, Y., Nozawa, M., Kawamura, K., Hirata, T., Kubota, Y., Sata, M., Kuwabara, K., Higashiyama, A., Kadota, A., Sugiyama, D., Miyamatsu, N., Miyamoto, Y., & Okamura, T. (2020). Estimated 24 h Urinary Sodium-to-Potassium Ratio Is Related to Renal Function Decline: A 6-Year Cohort Study of Japanese Urban Residents. International journal of environmental research and public health, 17(16), 5811. https://doi.org/10.3390/ijerph17165811

Response 13: We read through these two studies and added these two studies as references (Line 275-278) to support the association between sodium and kidney function decline.

Point 14: Check also this preprint from Srilanka.

https://www.medrxiv.org/content/10.1101/2020.04.17.20068833v1.full

Response 14: This study measured the salt, and potassium intake among the residents in Sri Lanka. Although this study is interesting, it is not related with the objective of our study. Therefore, we did not used this study for reference.

Round 2

Reviewer 1 Report

The authors have improved the manuscript. However, I still have minor suggestions.

In figure 2, the authors should provide which quintile presented a statistical difference when compared to another, not only provide the p value. Ex: Q1 vs Q5. It is not clear in the figure nor in the text. Even more in potassium box-plots, where it visually appears that the box plots were not different. 

Same suggestion for Figure 3a-c.

Author Response

Dear Reviewer,

Thank you for dedicating your time and efforts to reviewing our manuscript. We revised the manuscript according to your comments. Please see below for a point-by-point response to your comments.

Point 1: The authors have improved the manuscript. However, I still have minor suggestions.

In figure 2, the authors should provide which quintile presented a statistical difference when compared to another, not only provide the p value. Ex: Q1 vs Q5. It is not clear in the figure nor in the text. Even more in potassium box-plots, where it visually appears that the box plots were not different.

Response 1: The P value showed on Figure 2 was from the nonparametric equality-of-medians test. The null hypothesis of this test was that the median of five quintile groups were equal. Therefore, this test could not tell us which quintile presented a statistical difference when compared to another. To further delineate the difference between quintiles, we decided to perform the Kruskal Wallis test and examine whether the distribution of AER was different between each pair of quintile groups. All pairs of significant difference were showed by connected lines in the revised Figure 2. And relevant revision was made in the methods (Line125-127) and results section (Line 188-193).

Point 2: Same suggestion for Figure 3a-c.

Response 2:  Figure 3a-c presented the main results of the mixed-effects logistic regression models. In these models, the first quintile was used as the reference group. If the 95%CI of the adjusted ORs do not include “1”, it means that the odds of albuminuria increased/decreased significantly compared with the reference level (Q1). The P values on this figure were from the regression models indicated whether there was a statistically significant association between the quintiles of sodium, potassium, and Na/K ratio and the proportion of albuminuria after adjusting for covariates. We added a few sentences in the Result section (Line 242-247) to explain the P value showed in Figure 3A-C for clarification. “The P value in Figure 3A-C indicated whether there was a statistically significant association between the quintiles of sodium, potassium, and Na/K ratio and the proportion of albuminuria after adjusting for covariates in the logistic regression models. As showed in this figure, quintiles of sodium and Na/K ratio were significantly associated with the proportion of albuminuria in all three models, while quintiles of potassium were not found associated with the outcome variable in any models.”
